# Physiotherapy Within Inpatient Mental Health Wards: Involvement, Diagnoses, and Lifestyle Characteristics

**DOI:** 10.3390/healthcare13030279

**Published:** 2025-01-30

**Authors:** Philip Hodgson, Laura Hemmings, Brendon Stubbs, Davy Vancampfort, Erin Byrd

**Affiliations:** 1Physiotherapy Department, Tees, Esk and Wear Valleys NHS Foundation Trust, West Park Hospital, Edward Pease Way, Darlington DL2 2TS, UK; 2School of Science, Technology and Health, York St John University, Lord Mayor’s Walk, York YO31 7EX, UK; 3School of Sport, Exercise and Rehabilitation Sciences, University of Birmingham, Birmingham B15 2TT, UK; l.hemmings@bham.ac.uk; 4Centre for Sport Science and University Sports, University of Vienna, 1150 Vienna, Austria; brendon.stubbs@kcl.ac.uk; 5Department of Psychological Medicine, Institute of Psychiatry, Psychology and Neuroscience, King’s College London, De Crespigny Park, London SE5 8AF, UK; 6Department of Rehabilitation Sciences, KU Leuven, 3000 Leuven, Belgium; davy.vancampfort@kuleuven.be; 7School of Sport, Faculty of Life and Health Sciences, Ulster University, Derry-Londonderry BT48 7JL, UK; e.byrd@ulster.ac.uk

**Keywords:** physiotherapy, mental health, physical health, inpatient, lifestyle

## Abstract

Background: Severe mental illness (SMI) is often linked to physical health issues, including multiple comorbidities. Physiotherapists are increasingly recognized for their role in addressing these health disparities. This study investigated the role of physiotherapy in managing physical health conditions in individuals admitted to inpatient mental health services. Objective: The primary aim was to examine the prevalence of physical comorbidities among individuals admitted to inpatient mental health services, comparing those referred to physiotherapy versus those not referred. Secondary aims included assessing the relationship between physiotherapy referral and admission duration and identifying patterns in referral likelihood based on primary and comorbid diagnoses. Methods: A data linkage analysis was conducted using records from Tees, Esk and Wear Valleys NHS Foundation Trust, encompassing admissions from September 2020 to January 2023. Demographic data, physiotherapy referral status, and SNOMED-CT coded diagnoses were analyzed. Results: Among 2150 admissions, 505 (23.5%) were referred for physiotherapy. Multimorbidity was present in 58.1% of admissions, with a higher prevalence (67.8%) in those referred to physiotherapy versus those not referred (55.1%). Individuals referred to physiotherapy had longer lengths of stay (117.3 days), compared to those not referred (44.1 days), suggesting that extended stays may indirectly facilitate the identification and management of physiotherapy needs. Referral likelihood was influenced by primary diagnoses and comorbidities. Conclusions: Approximately one in four inpatient admissions resulted in a physiotherapy referral, with a higher prevalence of multimorbidity in those referred. Further research is warranted to explore the long-term impacts of physiotherapy interventions on physical and mental health outcomes.

## 1. Introduction

The term severe mental illness (SMI) is commonly used to refer to mental health disorders that often impact an individual’s ability to engage in personal, social, and occupational activities [1]. Examples of such conditions often include schizophrenia, schizoaffective disorders, bipolar disorder, and major depression [2].

It is well-recognized that there is a complex and bidirectional link between physical and mental health [3,4]. As many as one in three people who have a long-term physical health condition also experience symptoms of mental illness, most commonly depression or anxiety [5]. This figure increases to seven in ten people when exclusively considering those experiencing SMI [6]. It is therefore unsurprising that these significant health inequalities result in a reduced life expectancy of around 10–20 years [7]. This is particularly alarming, given that a large proportion of these premature deaths are attributed to potentially modifiable behaviors and are therefore preventable [8].

Recent data suggest that the average annual healthcare costs for a person with SMI is about GBP 5000 [9]. Moreover, costs for people presenting with comorbid physical and mental health needs are found to need approximately GBP 2000 more than those without SMI [10]. It is therefore reasonable to assume that any optimization in treatment is likely to result in reduced financial costs and a reduction in the need for unplanned intervention from clinical services.

Mental health settings can be obesogenic due to restrictions on movement, reduced access to outdoor space, increased access to unhealthy food, and less control over food choices [11]. Despite this, in the right circumstances, the inpatient setting can provide an opportunity for healthcare professionals to initiate and support people with SMI in making positive lifestyle changes or addressing unresolved health issues [12,13,14,15]. Such health promotion activities may include access to a qualified professional to deliver interventions, social support of peers and healthcare professionals, education, and personalized goal setting and activity plans [16]. Physiotherapists, as professionals who are able to appreciate mind–body interactions, are well-positioned to help address this clinical need [17]. Various physiotherapy bodies have provided guidelines and recommendations on the topic of the potential influence of physiotherapists in improving both the physical and mental health of people with SMI [18,19].

A recent review summarized existing research regarding physiotherapy in mental health settings [20], documenting advances in a field that has expanded substantially since 2015. This review found substantial evidence supporting exercise and physical activity interventions but reported some ambiguity around other physiotherapy interventions, identifying the importance of understanding referrals into physiotherapy within a UK mental health context. Furthermore, there was also variation across countries, highlighting the need to understand need and resource at a local level via economic evaluations of physiotherapy interventions and for more consumer-driven or patient experience studies.

Given the sparsity of published research considering inpatient mental health physiotherapy services, this analysis utilized an established dataset based on all admissions to inpatient adult mental health services across an area of Northeast England (1 September 2020–30 January 2023). The primary aim was to examine the prevalence of physical comorbidities among individuals admitted to inpatient mental health services, comparing those referred to physiotherapy versus those not referred. Secondary aims included assessing the relationship between physiotherapy referral and admission duration and identifying patterns in referral likelihood based on primary and comorbid diagnoses.

## 2. Materials and Methods

A data linkage analysis was completed using data from inpatient mental health services within Tees, Esk and Wear Valleys NHS Foundation Trust and was registered and approved by the Trusts Clinical Audit and Effectiveness Team (Project Number: 7144AMH22). The data linkage approach allowed the integration of multiple datasets, providing a comprehensive view of patient demographics, diagnoses, and physiotherapy referrals. This approach enabled the identification of trends in referral patterns and comorbidities, offering real-world insights into clinical practices.

A Microsoft Excel database containing all physiotherapy referrals was kept from the point of service inception (September 2020). This database contained information including site, ward, method of referral, reason for referral, date referral received, date first assessment attempted, discharge date from physiotherapy, and any onward referrals made by physiotherapy. In addition, the number of physiotherapy contacts per week was recorded for each of the eight wards covered across the three physical sites. Although data collection is ongoing, this manuscript considered inpatient admissions and physiotherapy referrals received until 30 January 2023.

This database was combined with routinely collected data provided by the Trust Business Intelligence & Clinical Outcomes Department, which provided additional anonymized information for all admissions to these wards during this time. Additional information included: date of admission, date of discharge, patient age, gender, ethnicity, primary diagnosis, and up to five comorbidities. Primary diagnosis and comorbidities were exported directly from clinical records, and in cases where more than five comorbidities were present, only the five highest ranked (according to SNOMED CT code and by date) were exported. Further information regarding body mass index (BMI), weight, diabetes, smoking, drug and alcohol use, and physical activity records was also collected.

The 209 unique primary diagnoses and 810 unique comorbidities retrieved were combined into 32 broader categories independently by two reviewers (PH and EB) and cross-checked. A third reviewer (LH) was available in case there was no consensus between both reviewers, which was not the case. The 32 categories are listed within Appendix A.

Initial primary diagnosis and comorbidities were then transferred to the corresponding group. The presence of each diagnosis/comorbidity was binary-coded for each admission (0 = not present, 1 = present). Admissions were divided into distinct groups of those referred to physiotherapy versus those not referred to physiotherapy to allow for analysis and comparison between groups.

### Data Analysis

Descriptive statistics are presented to provide an insight into population demographics, the length of physiotherapy involvement by diagnosis and comorbidities, length of admission for individuals referred to physiotherapy in comparison to individuals not referred during their admission, and the overall prevalence of the diagnostic or comorbidity categories between groups. As data did not meet normality assumptions, verified through the Shapiro–Wilk test, a Mann–Whitney U test was conducted to investigate the difference in overall length of admission for those referred to physiotherapy compared to those not referred. The Pearson chi-square test was used to assess relationships between categorical variables, such as referral reasons and diagnoses. Binary logistic regression was chosen to quantify associations between diagnoses and referral likelihood, providing odds ratios (ORs) and their respective 95% confidence intervals. All analyses were performed using IBM SPSS (version 29).

## 3. Results

There were, in total, 2150 admissions to adult mental health inpatient wards across three hospital sites and eight wards between 1 September 2020 and 30 January 2023. Of these admissions, 505 (23.5%) were referred for physiotherapy. Although the physiotherapy service covered inpatient settings only, due to a lack of community provision, four additional community referrals were made and were therefore included in our analysis. Demographic information is detailed within Table 1.

### 3.1. Physical Health of Individuals Admitted to Inpatient Mental Health Services

Across the entire cohort, 58.13% of individuals admitted presented with two or more physical health comorbidities alongside their primary diagnosis (Table 2). Of these admissions, individuals referred to physiotherapy tended to have a greater number of physical health comorbidities (with 67.78% having two or more physical health comorbidities). In comparison, physical multimorbidity was still common in individuals not referred to physiotherapy (with 55.13% having two or more physical health comorbidities); however, it was much lower than in those individuals who were referred to physiotherapy during their inpatient admission. From this data, it appears that individuals with 1–2 comorbidities were less likely to be referred to physiotherapy, with individuals with 3+ comorbidities being more likely to receive a physiotherapy referral during their inpatient admission.

### 3.2. Prevalence of Physical Comorbidities Between Individuals Referred to Physiotherapy Compared to Those with No Physiotherapy Involvement

Table 3 shows the prevalences of the various physical comorbidities for individuals referred to physiotherapy in comparison to individuals not referred to physiotherapy. In both groups, the most common physical comorbidity was related to acute medical issues, with the next most common being metabolic and endocrine issues and gastrointestinal conditions, regardless of whether a physiotherapy referral was made. Differences between groups indicated that the presence of fatigue syndromes, musculoskeletal-related conditions, neurological issues, or pain was more common in those referred to physiotherapy.

### 3.3. Length of Physiotherapy Involvement by Reason for Referral and Primary Diagnosis

Table 4 details the length of physiotherapy involvement by reason for referral. Overall, individuals referred for mobility concerns were the largest single group, followed by the various musculoskeletal (MSK) categories. Average length of physiotherapy involvement was 26.72 days; however, a large variation was included within this. On average, individuals referred for input relating to functional neurological disorders tended to spend the longest amount of time involved with physiotherapy during their admission (m = 246.60 days), with involvement for all other primary diagnoses ranging from 5 days to 40 days.

Table 5 details the length of physiotherapy involvement by primary diagnosis. Individuals with a neurodevelopmental primary diagnosis recorded tended to have the longest average physiotherapy involvement (m = 120.36 days), with involvement for all other primary diagnoses ranging from 8 days to 36 days.

### 3.4. Length of Admission (Physiotherapy Involvement Compared to Those Not Referred to Physiotherapy)

Table 6 displays the average length of admission by diagnosis for all admissions during the selected timeframe. Overall, individuals referred to physiotherapy generally had a longer admission (117 days) in comparison to individuals not referred to physiotherapy (44 days) (U = 549,263.50, *p* < 0.001).

### 3.5. Relationship Between Primary or Comorbid Diagnoses and Reason for Referral to Physiotherapy

Several statistically significant relationships between primary and comorbid diagnoses and reason for referral to physiotherapy were determined through chi-square analysis (Table 7). The strongest relationships between reason for referral and primary diagnosis were for schizophrenia and MSK extremity (X^2^ = 121.06); PTSD and depressive disorders; and Ministry of Defense (MOD) protocol (X^2^ = 25.53 and 29.44 respectively). The strongest relationships between comorbidities and reason for referral were between neurological issues and respiratory referrals (X^2^ = 194.91); neurological issues and neurology referrals (X^2^ = 113.09); and metabolic and endocrine conditions and mobility referrals (X^2^ = 74.25).

### 3.6. Likelihood of Referral to Physiotherapy by Primary Diagnosis and Comorbidities

Unadjusted binary logistic regression examined the association between primary diagnoses and the likelihood of a person receiving a referral to physiotherapy services. People with eating disorders had 1.60-fold increased odds (95% CI: 0.70–3.65) of being referred to physiotherapy, schizophrenia had 1.61-fold increased odds of being referred (95% CI: 0.75–3.43), obsessive–compulsive and hypochondriacal conditions had 1.98-fold increased odds of being referred (95% CI: 0.40–9.77), organic and neurodegenerative conditions were at 2.40-fold increased odds (95% CI: 0.76–7.60), and somatoform conditions had 3.30-fold increased odds of being referred to physiotherapy (95% CI: 0.57–18.99). However, none were statistically significant.

Similarly, unadjusted binary logistic regression examined the association between physical health comorbid diagnoses and the likelihood of a person being referred to physiotherapy. People presenting with fatigue as a comorbid condition were at 6.38-fold increased odds of receiving a physiotherapy referral (CI: 1.41–28.94, *p* = 0.016), and people with musculoskeletal-related conditions were at 3.48-fold increased odds (CI: 1.23–9.81, *p* = 0.019) of receiving a physiotherapy referral.

## 4. Discussion

This study highlights the prevalence of multimorbidity in patients admitted to inpatient mental health services and examines patterns of physiotherapy referral based on primary and comorbid diagnoses. By doing so, the current study aims to provide insight into the role of physiotherapists within inpatient mental health services, identify gaps in referrals, and advocate for the better integration of physiotherapy services. The findings are intended to serve as a blueprint for budget holders, policy makers, and other health professionals proposing or establishing a physiotherapy service within inpatient adult mental health settings. Despite physiotherapists having the potential to enhance the identification and treatment of physical health issues [18,21], recent literature indicates a limited understanding of the physiotherapists’ role in mental health among multidisciplinary teams and service providers [22,23]. This analysis seeks to raise awareness of the value of physiotherapy, promote existing services, and support the development of new ones where necessary.

The finding that 58.1% of admissions involved two or more physical health comorbidities aligns with evidence of high physical co-morbidity in individuals with SMI, underscoring the complexity of patient presentations within this setting [24]. This figure is greater than the prevalence of 25% reported within a recent systematic review and meta-analysis [25]; however, the meta-analysis does not provide a direct comparison to inpatient populations. Furthermore, our data may demonstrate the above-average level of health inequality experienced in this area of the UK [26]. While individuals referred to physiotherapy had more comorbidities than those not referred, many with multiple comorbidities still did not receive referrals.

Barriers to optimized referral pathways, such as insufficient integration of physiotherapists within mental health multidisciplinary teams, suggest missed opportunities to address physical health issues [23]. Additionally, patient experience research highlights insufficient attention to physical health in mental health settings [27], further limiting referrals to physiotherapy. This is concerning, as it reflects systematic barriers to physiotherapy, despite service availability. These specific barriers may include limited staff and patient awareness of the benefits of physiotherapy, leading to underutilization and resource constraints like staffing shortages that restrict the identification of need and subsequent referral to physiotherapy services. Stigma surrounding SMI may deter patients from seeking physiotherapy, while communication gaps among healthcare providers is likely to hinder effective referrals, resulting in missed opportunities for improving patient outcomes.

The most common physical comorbidities identified—acute medical issues, metabolic and endocrine disorders, and gastrointestinal conditions—align with previous research [28,29], confirming the representativeness of our sample. Patients presenting with fatigue, schizoaffective disorders, neurological complaints, and musculoskeletal conditions had the highest likelihood of referral to physiotherapy, reflecting staff recognition of the benefits of physiotherapy in these areas [30,31,32,33]. Musculoskeletal, neurological, and cardiorespiratory medicine are core practice areas for physiotherapists in the UK [34], making it encouraging that patients with neurological and musculoskeletal symptoms are amongst the most likely to be referred. On the other hand, only 40.4% of patients with musculoskeletal conditions, 41.7% with neurological conditions, and 30.0% of respiratory patients were referred to physiotherapy, highlighting missed opportunities for intervention and the need for staff education to broaden awareness of physiotherapy’s scope. It is important to highlight that different comorbidities are likely to result in distinct implications for physiotherapy involvement. For example, musculoskeletal conditions may require pain- or mobility-focused interventions [35,36], while neurological issues may require tailored approaches to manage functional impairments [37].

Mobility concerns as a single indication accounted for the largest proportion of physiotherapy referrals (24.8%), suggesting a significant level of frailty within the patient population, despite the service being designed for working-age adults (aged 18–66). This aligns with evidence that people with SMI have a higher frailty prevalence compared to the general population [38]. Combined musculoskeletal issues comprised 41.3% of referrals, likely reflecting greater familiarity among staff with this aspect of physiotherapy in working-age adults.

Despite physiotherapists’ capacity to promote lifestyle interventions such as physical activity, which positively affect both mental and physical health [39], referrals for lifestyle advice account for less than 5% of referrals. This indicates underutilization of physiotherapists in promoting health-enhancing behaviors. Effective health promotion, as advocated by the Lancet Psychiatry Commission [3], could reduce the future burden of disease on individuals and improve patient outcomes.

It is particularly concerning that, despite 33% of individuals with SMI experiencing chronic pain [40], only one referral was specifically for pain management. Non-pharmacological interventions should be prioritized in pain management [41], yet current referral practices reveal a disconnect between service availability and utilization. Given the complex relationship between chronic pain and mental health, physiotherapists should be integrated into the diagnostic and management processes, reducing the reliance on medication and addressing chronic pain more holistically [42].

A remarkable observation was that our data suggested a potential correlation between increased physical health comorbidities and longer inpatient stays. Longer stays may provide opportunities for physical health issues to be identified, increasing the likelihood of physiotherapy referrals. Patients referred to physiotherapy had an average stay of 117 days, compared to 44 days for patients not referred. However, the average physiotherapy involvement of 27 days suggests that while physiotherapy is often initiated, referrals are delayed by an average of 91 days. Early identification and referral to physiotherapy are crucial for optimizing patient outcomes, and strategies for earlier referrals could benefit more patients.

Statistically significant relationships were identified between referral reasons and primary or comorbid diagnoses. For example, the association between schizophrenia and musculoskeletal injury aligns with evidence of lower bone mineral density [43], insufficient care for osteoporosis [44], and a higher risk for frailty and falls [45]. Additionally, the relationships between a patient referred on a MOD protocol and a primary diagnosis of depressive disorders or PTSD are supported by many studies, demonstrating a higher prevalence of both diagnoses in active or veteran military populations [46,47,48,49,50].

### 4.1. Limitations and Future Research

Whilst efforts were made to minimize limitations, reliance on clinical data from a single service reduces the generalizability of the findings to other settings. The absence of pre-/post-physiotherapy outcome data limits the ability to assess intervention effectiveness. Variability in the data reflects the diverse physical health needs of individuals with SMI, complicating the identification of clear patterns. Finally, reliance on referral data may not fully capture the demand for physiotherapy services, due to clinician biases or resource constraints.

Despite these limitations, this study provides the first comprehensive exploration of routinely collected data in inpatient mental health physiotherapy services. The findings offer valuable insights for improving service delivery and guiding future service development.

### 4.2. Recommendations

Inpatient admissions present an opportunity to address physical health comorbidities of individuals with SMI who are disproportionately affected by these issues [51,52] yet are less likely to engage with physical healthcare providers to address these issues [53]. By improving access to physiotherapy services during inpatient stays and providing post-discharge guidance, patients’ engagement with physical healthcare can be enhanced [23,27,54]. Without inpatient referrals, disparities in healthcare access may persist.

To optimize physiotherapy services within mental health settings, we suggest that healthcare providers and policymakers prioritize staff training on physiotherapy and the benefits of addressing physical health comorbidities. This should help to promote the early identification of issues, which should then be addressed via clear referral pathways and adequately resourced physiotherapy services integrated within multi-disciplinary teams. Within patient interactions, physiotherapists should incorporate lifestyle advice to address health inequalities and enhance care quality. Additionally, we recommend ongoing research to evaluate the long-term impacts of physiotherapy on physical and mental health outcomes. These recommendations aim to provide a practical framework for improving physiotherapy integration, ultimately benefiting individuals with SMI.

## 5. Conclusions

This study provides the first comprehensive exploration of physiotherapy referrals within inpatient mental health settings, offering practical insights for clinicians and administrators. The findings underscore the importance of timely physiotherapy referral and intervention in addressing the interplay between physical and mental health, improving patient outcomes, and reducing healthcare disparities. Future research building on these findings will be essential for refining care models and advancing physiotherapy’s role in mental health services.

## Figures and Tables

**Table 1 healthcare-13-00279-t001:** Demographics.

Characteristic	Referred to Physiotherapy	Not referred to Physiotherapy	All Admissions
Age	42.93 (14.56)	44.40 (13.21)	40.68 (13.59)
Gender	253 Male	862 Male	1115 Male
	252 Female	763 Female	1015 Female
	4 Other	20 Other	24 Other
Length of admission (days)	117.31 (237.45)	44.09 (87.69)	61.39 (142.07)
Ward Type	375 Acute	1396 Acute	1771 Acute
	63 Rehab	4 Rehab	67 Rehab
	28 PICU	176 PICU	204 PICU
	39 Eating disorders	69 Eating disorders	108 Eating disorders
	4 Community (no provision)	-	4 Community (no provision)

Data are presented as mean and standard deviation where appropriate or by frequency for categorical variables. PICU = psychiatric intensive care unit.

**Table 2 healthcare-13-00279-t002:** Number of physical health comorbidities.

Number of Physical Health Comorbidities	Referred to Physiotherapy *n* (%)	Not Referred to Physiotherapy *n* (%)	All Admissions *n* (%)
0	65 (12.77)	332 (20.18)	397 (18.43)
1	99 (19.45)	406 (24.68)	505 (23.44)
2	121 (23.77)	445 (27.05)	566 (26.28)
3	108 (21.22)	270 (16.41)	378 (17.55)
4	73 (14.34)	139 (8.45)	212 (9.84)
5	43 (8.45)	53 (3.22)	96 (4.46)
Mean (SD)	2.30 (1.47)	1.78 (1.34)	1.90 (1.39)

Data are presented as frequency and percentage, followed by mean and standard deviation, for each group.

**Table 3 healthcare-13-00279-t003:** Prevalence of physical health comorbidities in individuals presenting to physiotherapy services vs. not referred to physiotherapy.

Comorbid Diagnostic Grouping	Overall	Referred to Physiotherapy *n* (% Group)	Not Referred to Physiotherapy *n* (% Group)	LR Chi-Square	*p*-Value
Acute medical issues	1175	260 (22.13)	915 (77.87)	9.14	0.058
Cancer	35	0 (0)	35 (100)	16.31	<0.001 **
Cardiovascular conditions	408	119 (29.17)	289 (70.83)	9.93	0.042 *
Dermatology issues	162	48 (29.63)	114 (70.37)	9.09	0.028 *
Fatigue syndrome	13	7 (53.85)	6 (46.15)	5.53	0.019 *
Gastrointestinal conditions	358	125 (34.92)	233 (65.08)	18.93	<0.001 **
Lymphatic, rheumatic, and immunological disorders	44	9 (20.45)	35 (79.55)	1.11	0.574
Metabolic and endocrine disorders	522	162 (31.03)	360 (68.97)	16.70	0.002 *
Musculoskeletal (MSK)-related conditions	270	109 (40.37)	161 (59.63)	32.66	<0.001 **
Neurodevelopmental disorders	213	61 (28.64)	152 (71.36)	5.60	0.133
Neurological disorders	127	53 (41.73)	74 (58.27)	18.45	<0.001 **
Organic and neurodegenerative disorders	115	30 (26.01)	85 (73.99)	1.80	0.407
Pain	130	44 (33.85)	86 (66.15)	7.17	0.028 *
Reproductive conditions	56	10 (17.86)	46 (82.14)	1.39	0.499
Respiratory disorders	318	89 (27.99)	229 (72.01)	7.97	0.047 *
Somatoform-related disorders	3	1 (33.33)	2 (66.66)	0.15	0.704
Other physical health diagnosis	150	45 (30.00)	105 (70.00)	3.23	0.199

Data are presented as frequency and percentage; LR = likelihood ratio; * significant when *p* < 0.05, ** *p* < 0.001.

**Table 4 healthcare-13-00279-t004:** Length of physiotherapy involvement (days) by reason for referral.

Reason for Referral Grouping	Number of Referrals	Total Time on Physio Caseload (Days)	Average Length of Physio Involvement (Days)	SD (Days)
Chronic pain	1	6.00	6.00	N/A
Equipment	2	10.00	5.00	5.66
Functional neurological disorder	5	1233.00	246.60	506.75
Joint hypermobility	4	43.00	10.75	2.99
Lifestyle advice	24	959.00	39.96	48.82
Mobility	126	4406.00	34.97	55.46
MOD protocol	51	221.00	4.33	9.21
MSK—Extremities	102	2769.00	27.15	40.92
MSK—Multiple/Other	19	600.00	31.58	41.29
MSK—Spinal/Back pain	89	1901.00	21.36	35.54
Neuro	17	476.00	28.00	42.64
No information or discharged before assessment	42	42.00	1.00	0.00
Orthopedic/Trauma	23	801.00	34.83	31.64
Other	3	116.00	38.67	34.70
Respiratory	1	15.00	15.00	N/A
Overall	509	13,598.00	26.72	65.07

MSK = Musculoskeletal; MOD = Ministry of Defence; N/A = Not applicable.

**Table 5 healthcare-13-00279-t005:** Length of physiotherapy involvement (days) by primary diagnosis.

Primary Diagnosis Grouping	Number of Referrals	Total Time on Physio Caseload (Days)	Average Length of Physio Involvement (Days)	SD (Days)
Anxiety disorders	1	8.00	8.00	N/A
Bipolar disorders	50	1442.00	28.84	45.58
Depressive and other affective disorders	75	2089.00	27.85	61.98
Drug and substance use/misuse	31	368.00	11.87	19.98
Eating disorders	32	844.00	26.38	37.92
Mania	2	24.00	12.00	0.00
Neurodevelopmental disorders	11	1324.00	120.36	343.90
Obsessive–compulsive or hypochondriacal problems	3	11.00	3.67	4.62
Organic and neurodegenerative disorders	8	261.00	32.63	27.67
Other non-mental health diagnosis	15	542.00	36.13	56.30
Personality disorders	115	2562.00	22.28	36.11
Psychosis-related disorders	34	672.00	19.76	29.99
PTSD and trauma	18	247.00	13.72	21.82
Schizoaffective disorders	21	615.00	29.29	38.45
Schizophrenia disorders	78	2506.00	32.13	41.79
Self-harm	2	3.00	1.50	0.71
Somatoform-related disorders	3	17.00	5.67	8.08
Stress disorders	10	63.00	6.30	11.47
Total	509	13,598.00	26.72	65.07

PTSD = post-traumatic stress disorder; N/A = Not applicable.

**Table 6 healthcare-13-00279-t006:** Length of admission (days) by primary diagnosis.

	Overall			Referred to Physio			Not Referred to Physio			*p*-Value
	Total Admission Days	Mean	Standard Deviation	Total Admission Days	Mean	Standard Deviation	Total Admission Days	Mean	Standard Deviation	
Anxiety Disorders	948	37.92	78.35	53	53	N/A	895	37.29	79.97	0.165
Bipolar Disorders	14,940	75.45	117.61	7483	149.66	191.16	7457	50.39	62.04	<0.001 *
Depressive and other affective disorders	15,958	50.34	105.96	9756	130.08	188.4	6202	25.63	35.07	<0.001 *
Drug and substance use/misuse	6829	27.21	62.73	1414	45.61	87.67	5415	24.61	58.17	0.016 *
Eating Disorders	8175	83.42	55.48	3037	94.91	63.61	5138	77.85	50.67	0.303
Mania	562	35.13	38.1	68	34.00	25.46	494	35.29	40.31	0.700
Neurodevelopmental	13,610	261.73	573.03	7216	656	1081.08	6394	155.95	268.83	0.049 *
Neurological Issues	45	22.50	20.51	0	N/A	N/A	45	22.50	20.51	N/A
Obsessive–compulsive or hypochondriacal	166	20.75	28.35	26	8.67	5.86	140	28.00	34.84	0.393
Organic and neurodegenerative disorders	1352	71.16	76.72	784	98.00	89.28	568	51.64	63.41	0.091
Other mental health diagnosis	165	10.31	9.24	0	N/A	N/A	165	10.31	9.24	N/A
Other non-mental health diagnosis	12	6.00	2.83	0	N/A	N/A	12	6.00	2.83	N/A
Personality Disorders	13,942	31.54	72.7	6336	55.10	119	7606	23.26	43.98	0.016 *
Psychosis-related disorders	12,939	56.01	83.71	2924	86.00	89.53	10,015	50.84	81.79	<0.001 *
PTSD and Trauma	2799	49.98	125.68	1150	63.89	167.73	1649	43.39	102.05	0.150
Schizoaffective disorders	7079	91.94	109.43	3524	167.81	160.09	3555	63.48	64.85	0.017 *
Schizophrenia Disorders	31,780	133.53	188.71	15,745	201.86	238.07	16,035	100.22	149.08	0.002 *
Self-harm	121	7.56	5.64	17	8.50	7.78	104	7.43	5.65	0.817
Somatoform-related disorders	75	12.50	14.04	68	22.67	13.32	7	2.33	2.31	0.100
Stress Disorders	735	17.09	46.46	110	11.00	10.87	625	18.94	52.77	0.899
Overall	132,232	61.39	142.07	59,711	117.31	237.45	72,521	44.09	87.67	<0.001 *

PTSD = post-traumatic stress disorder; N/A = Not applicable; * = significant when *p* < 0.05.

**Table 7 healthcare-13-00279-t007:** Correlations between reason for physiotherapy referral and primary diagnosis and comorbidities.

Referral Reason	Primary Diagnosis	Chi-Square Correlation (Χ^2^)	*p*-Value	Comorbidities	Chi-Square Correlation (Χ^2^)	*p*-Value
Chronic Pain	Bipolar Disorders	9.88	0.002 *	Dermatological Issues	15.97	0.001 **
				MSK-Related Conditions	9.36	0.025 *
Equipment				MSK-Related Conditions	46.30	<0.001 **
Functional Neurological Disorder	Neurodevelopmental Disorders	6.58	0.01 *	Gastrointestinal Conditions	24.75	<0.001 **
	Personality Disorders	4.79	0.029 *	MSK-related Conditions	66.11	<0.001 **
				Neurodevelopment Disorders	8.25	0.041 *
				Neurological Issues	37.63	<0.001 **
Joint Hypermobility	Personality Disorders	7.29	0.007 *	MSK-Related Conditions	19.62	<0.001 **
Lifestyle Advice	Schizophrenia Disorders	23.15	<0.001 **	Dermatological Issues	29.868	<0.001 **
				Fatigue Syndrome	5.14	0.023 *
				Gastrointestinal Conditions	8.82	0.032 *
				Other	10.16	0.006 *
Mobility	Bipolar Disorders	13.17	<0.001 **	Acute Medical Issues	12.82	0.012 *
	Drug and Substance Misuse	6.17	0.013 *	Bipolar Disorders	9.99	0.002 *
	Organic and Neurodegenerative Disorders	8.05	0.005 *	Cardiovascular Conditions	32.89	<0.001 **
	Psychosis-Related Disorders	4.97	0.026 *	Drug and Substance Misuse	15.44	0.009 *
	Schizophrenia Disorders	12.51	<0.001 **	Fatigue Syndrome	14.75	<0.001 **
				Gastrointestinal Conditions	59.86	<0.001 **
				Metabolic and Endocrine Issues	74.25	<0.001 **
				MSK-Related Conditions	37.88	<0.001 **
				Other	14.87	<0.001 **
				PTSD and Trauma	26.05	<0.001 **
				Respiratory Issues	29.92	<0.001 **
				Self-Harm	14.06	0.007 *
				Somatoform-Related Disorders	4.12	0.042 *
MOD Protocol	Depressive and Other Affective Disorders	29.44	<0.001 **	Acute Medical Issues	12.89	0.012 *
	Psychosis-Related Disorders	6.28	0.012 *	Respiratory Issues	8.16	0.043 *
	PTSD and Trauma	25.53	<0.001 **	Self-Harm	11.57	0.021 *
	Schizophrenia Disorders	6.49	0.011 *			
	Stress Disorders	9.13	0.003 *			
MSK Extremity	Depressive and Other Affective Disorders	8.22	0.004 *	Acute Medical Issues	21.81	<0.001 **
	Drug and Substance Misuse	7.89	0.005 *	Depressive and Other Affective Disorders	10.11	0.006 *
	Schizophrenia Disorders	121.06	<0.001 **	Neurodevelopment Disorders	8.19	0.042 *
				Other Mental Health Diagnosis	7.95	0.019 *
				Personality Disorders	5.98	0.05 *
				Respiratory Issues	11.67	0.009 *
MSK Multiple/Other	Schizoaffective Disorders	16.99	<0.001 **	Pain	8.06	0.018 *
				Personality Disorders	11.56	0.003 *
				Respiratory Issues	10.58	0.014 *
MSK Spine	Eating Disorders	4.21	0.04 *	Anxiety Disorders	7.60	0.022 *
	Psychosis-Related Disorders	5.24	0.022 *	Metabolic and Endocrine Issues	19.36	<0.001 **
	Somatoform-Related Disorders	12.95	<0.001 **	MSK-Related Conditions	60.95	<0.001 **
				Neurological Issues	12.00	0.002 *
				Obsessive–Compulsive or Hypochondriacal Disorders	11.04	0.004 *
				Pain	20.61	<0.001 **
Neurological	Organic and Neurodegenerative Disorders	23.21	<0.001 **	MSK-Related Conditions	7.81	0.05 *
				Neurological Issues	113.09	<0.001 **
				Organic and Neurodegenerative Disorders	31.86	<0.001 **
Orthopedic or Trauma	Neurodevelopmental Disorders	3.89	0.048 *	Dermatological Issues	29.81	<0.001 **
	Schizophrenia Disorders	5.35	0.021 *	MSK-Related Conditions	11.55	0.009 *
				Other	38.53	<0.001 **
				Personality Disorders	6.74	0.034 *
				Schizoaffective Disorders	18.56	<0.001 **
				Self-Harm	29.83	<0.001 **
Other Reason for Referral	Schizoaffective Disorders	7.72	0.005 *	Lymphatic, Rheumatic, and Immunological Issues	16.33	<0.001 **
				Sensory Dysfunction	14.70	<0.001 **
Respiratory	Eating Disorders	20.99	<0.001 **	Depressive and Other Affective Disorders	7.98	0.019 *
				Lymphatic, Rheumatic, and Immunological Issues	52.88	<0.001 **
				Neurodevelopment Disorders	13.09	0.004 *
				Neurological Issues	194.91	<0.001 **

MSK = Musculoskeletal; PTSD = post-traumatic stress disorder; * significant when *p* < 0.05. ** *p* < 0.001.

## Data Availability

The raw data supporting the conclusions of this article will be made available by the authors upon request.

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
