# Peer review of "Physiotherapy Within Inpatient Mental Health Wards: Involvement, Diagnoses, and Lifestyle Characteristics"

_healthcare, 2025, doi:10.3390/healthcare13030279_

Round 1
Reviewer 1 Report
Comments and Suggestions for Authors
The manuscript (healthcare-3403322) explores about the contribution of physiotherapy in inpatient mental health wards, with respect to the prevalence of comborbidities, referral patterns, and the interdependence between intervention of physiotherapy and staying duration. The outcomes of the study are quite relevatnt and important, however, there are several areas which needs to be clarified and improved, as mentioned as below:
1. Authors should include identification of key comorbidities and their physiotherapy related consequences.
2. There are lot of inconsistencies with regards to use of terminologies, e.g. the use of phrases like “individuals with SMI” or “patients with mental health disorders” may confuse the readers.
3. Authors should discuss the rationale for selecting data linkage analysis.
4. The manuscript does not include a comprehensive description about particular barriers, like personal awareness, limitations in resources, etc. for providing more insights to enhacen referral pathways.
5. Whether the analysis regarding post-physiotherapy outcomes like, functional enhacnements, life quality, etc. were done. If not, the limitations should be clearly discussed.
6. The discussions should be supported by comparison of outcomes with previous research, especially with resepect to prevalence of comorbidities and effect of physiotherapy on mental health.
7. Authors should provide clarifications regarding criteria for chosing the statistical tests in the study.
8. The conclusion may also include the suggestions for policy makers to enhance incorporation of physiotherapy services.
Author Response
Reviewer: 1
Comment 1: Authors should include identification of key comorbidities and their physiotherapy related consequences.
Response: We appreciate the reviewer’s suggestion to include key comorbidities and their physiotherapy-related consequences. In response, we would like to highlight a section in our discussion detailing the most prevalent comorbidities—acute medical issues, metabolic/endocrine disorders, and gastrointestinal conditions (Lines 267-272). Following your suggestion, we have expanded on this to include information regarding the implications for physiotherapy (Lines 278-282). For example, musculoskeletal conditions may require pain or mobility-focused interventions, while neurological issues may require tailored approaches to manage functional impairments. These additions emphasise the diverse role of physiotherapy in addressing complex physical health needs within mental health inpatient settings. Thank you for this valuable feedback, which we feel has strengthened our manuscript.
Comment 2: There are lot of inconsistencies with regards to use of terminologies, e.g. the use of phrases like “individuals with SMI” or “patients with mental health disorders” may confuse the readers.
Response: We appreciate the reviewer’s feedback regarding the inconsistencies in terminology. To address this, we have carefully reviewed the manuscript and standardised our terminology throughout. We now use consistent terminology to ensure clarity and avoid confusion for readers. Thank you for your valuable suggestion.
Comment 3: Authors should discuss the rationale for selecting data linkage analysis.
Response: We thank the reviewer for highlighting the need to discuss the rationale for selecting data linkage analysis. In response, we have clarified this in the manuscript (Lines 92-95). Data linkage analysis was chosen as it allowed integration of multiple datasets, providing a comprehensive view of patient demographics, diagnoses, and physiotherapy referrals. This approach enabled the identification of trends in referral patterns and comorbidities, offering real-world insights into clinical practices. These factors made data linkage the most suitable method for our study objectives. Thank you for this valuable suggestion.
Comment 4: The manuscript does not include a comprehensive description about particular barriers, like personal awareness, limitations in resources, etc. for providing more insights to enhance referral pathways.
Response: We thank the reviewer for highlighting the need to discuss barriers such as personal awareness and resource limitations that impact referral pathways. In response, we have added a section addressing these barriers, including limited patient awareness of physiotherapy benefits, resource constraints like staffing shortages, and stigma affecting perceptions of physiotherapy. We also discuss communication gaps between healthcare providers that hinder effective referrals. These additions (Lines 256-266) aim to provide deeper insights into improving referral pathways. Thank you for this valuable suggestion.
Comment 5: Whether the analysis regarding post-physiotherapy outcomes like, functional enhancements, life quality, etc. were done. If not, the limitations should be clearly discussed.
Response: We thank the reviewer for raising the importance of analysing post-physiotherapy outcomes such as functional enhancements and quality of life. As our study primarily focused on referral patterns and comorbidities, we did not assess these outcomes. We have explicitly acknowledged this as a limitation in the manuscript (Lines 321-323) and highlighted the need for future research to explore the long-term impacts of physiotherapy on functional and quality-of-life outcomes in this population (Lines 344-347). Thank you for your valuable suggestion.
Comment 6: The discussions should be supported by comparison of outcomes with previous research, especially with respect to prevalence of comorbidities and effect of physiotherapy on mental health.
Response: We thank the reviewer for suggesting the inclusion of comparisons with previous research regarding the prevalence of comorbidities and the effects of physiotherapy on mental health. In response, we have expanded the discussion to include relevant studies (Lines 249-253). For example, our finding of 58.1% physical multimorbidity is greater than the prevalence of 25% reported within a recent systematic review and meta-analysis, however, the meta-analysis does not provide a direct comparison to inpatient populations. Furthermore, our data may demonstrate the above-average level of health inequality experienced in this area of the UK. Additionally, we reference evidence supporting the role of physiotherapy within inpatient mental health settings, contextualising our results within the broader literature. These additions strengthen the manuscript by providing a more comprehensive perspective. Thank you for this valuable feedback.
Comment 7: Authors should provide clarifications regarding criteria for choosing the statistical tests in the study.
Response: We appreciate the reviewer’s request for clarification on the criteria for selecting statistical tests in our study. In response, we have added a concise explanation in the methods section (Lines 128-141). This states that as data did not meet normality assumptions, verified through the Shapiro-Wilk test, a Mann-Whitney U test was conducted to investigate the difference in overall length of admission for those referred to physiotherapy compared to those not referred. The Pearson chi-square test was used to assess relationships between categorical variables, such as referral reasons and diagnoses. Binary logistic regression was chosen to quantify associations between diagnoses and referral likelihood, providing odds ratios (OR’s) and their respective 95% confidence intervals. We hope that this explanation helps to clarify the rationale for choosing our specific statistical tests. Thank you for your valuable feedback.
Comment 8: The conclusion may also include the suggestions for policy makers to enhance incorporation of physiotherapy services.
Response: We appreciate the reviewer’s suggestion to include recommendations for policymakers to enhance the incorporation of physiotherapy services in our conclusion. In response, we have added specific recommendations (Lines 338-347). We suggest that healthcare providers and policymakers prioritise staff training on physiotherapy and the benefits of addressing physical health comorbidities. This should help to promote the early identification of issues, which should then be addressed via clear referral pathways and adequately resourced physiotherapy services integrated within multi-disciplinary teams. Within patient interactions, physiotherapists should incorporate lifestyle advice to address health inequalities and enhance care quality. Additionally, we recommend ongoing research to evaluate the long-term impacts of physiotherapy on physical and mental health outcomes. These recommendations aim to provide a practical framework for improving physiotherapy integration, ultimately benefiting individuals with SMI. Thank you for your valuable feedback.
Reviewer 2 Report
Comments and Suggestions for Authors
Many thanks for the opportunity to review an article related to Physiotherapy within Inpatient Mental Health Wards. The topic of scientific paper raised is very important nowadays. Therefore, I congratulate the authors on the idea and implementation of carrying out scientific research related to the topic of the paper presented above. However, I would like to make a few comments below, which I am sure will enhance the scientific and substantive quality of the submitted article. The first part of the research paper sets out very meticulously and factually what the aims of the research paper are. However, it makes sense to add a more detailed practical conclusion for the future. The introduction to the topic of the research paper is of a high standard, the authors have used many scientific publications related to the topic of the research paper. I have no objections to the introduction.
In the Data Analysis section, there is a lack of explanation about the selection of test controls. Information and justification for test selection, non-parametric normal/non-normal distribution should be added. There is a lack of explanation of the abbreviations in the table under all tables. Please add such information. The p-value should be written in lower case. In table number 3 there is a value (70.00%) but no numerical value is available. In the conclusions section, please expand the information on practical conclusions regarding care models and advancing physiotherapy's role in mental health services. A final comment concerns citations. Please standardise the naming of citations in a scientific article.
Author Response
Reviewer: 2
Comment 1: The first part of the research paper sets out very meticulously and factually what the aims of the research paper are. However, it makes sense to add a more detailed practical conclusion for the future. The introduction to the topic of the research paper is of a high standard, the authors have used many scientific publications related to the topic of the research paper. I have no objections to the introduction.
Response: We appreciate the reviewer’s positive feedback on the introduction and the presentation of our research aims. In response to the suggestion for a more detailed practical conclusion, we have enhanced this section within our discussion to include actionable recommendations for future practice (Lines 338-347). Our revised conclusion emphasises the need for integrating physiotherapy services within inpatient mental health settings. We suggest that healthcare providers and policymakers prioritise staff training on physiotherapy and the benefits of addressing physical health comorbidities. This should help to promote the early identification of issues, which should then be addressed via clear referral pathways and adequately resourced physiotherapy services integrated within multi-disciplinary teams. Within patient interactions, physiotherapists should incorporate lifestyle advice to address health inequalities and enhance care quality. Additionally, we recommend ongoing research to evaluate the long-term impacts of physiotherapy on physical and mental health outcomes. These recommendations aim to provide a practical framework for improving physiotherapy integration, ultimately benefiting individuals with SMI. Thank you for your valuable feedback.
Comment 2: In the Data Analysis section, there is a lack of explanation about the selection of test controls. Information and justification for test selection, non-parametric normal/non-normal distribution should be added.
Response: We thank the reviewer for highlighting the need to explain the selection of test controls and statistical methods in the Data Analysis section. In response, we have added further clarification to the manuscript (Lines 128-141). The selection of statistical tests was based on data characteristics and distribution. The Mann-Whitney U test was used for comparing admission lengths due to non-normal data distribution, verified through the Shapiro-Wilk test. Pearson’s chi-square test was applied to assess relationships between categorical variables, such as referral reasons and diagnoses. Binary logistic regression was chosen to quantify associations between diagnoses and referral likelihood, providing odds ratios and confidence intervals. These justifications ensure that our methods align with the study objectives and data properties. Thank you for your valuable feedback.
Comment 3: There is a lack of explanation of the abbreviations in the table under all tables. Please add such information.
Response: We appreciate the reviewer’s comment regarding the need for explanations of abbreviations used in the tables. In response, we have added a comprehensive list of abbreviations below each table to ensure clarity for readers. This addition will help enhance understanding and accessibility of the data presented. Thank you for your valuable feedback, which has improved the manuscript's clarity.
Comment 4: The p-value should be written in lower case.
Response: We appreciate the reviewer’s comment regarding the formatting of p-values. In response, we have revised the manuscript to ensure that all instances of "p-value" are written in lowercase as "p-value” or “p”. This change has been applied consistently throughout the document to enhance clarity and adherence to formatting standards. Thank you for your valuable feedback.
Comment 5: In table number 3 there is a value (70.00%) but no numerical value is available.
Response: We appreciate the reviewer’s observation regarding the missing numerical value associated with the 70.00% figure in Table 3. In response, we have reviewed the table and added the necessary numerical value to ensure clarity and completeness. Thank you for your valuable feedback, which has helped improve the accuracy of our manuscript.
Comment 6: In the conclusions section, please expand the information on practical conclusions regarding care models and advancing physiotherapy's role in mental health services.
Response: We thank the reviewer for highlighting the need to expand the conclusions regarding care models and advancing the role of physiotherapy in mental health services. In response, we have revised the conclusion to emphasise the integration of physiotherapy into multidisciplinary care models within mental health settings (Lines 338-347). We recommend structured referral pathways, increased staff training on physiotherapy benefits, and resource allocation to support these services. We also recommend further research to evaluate the long-term impacts of physiotherapy on patient outcomes, supporting its broader implementation in mental health care. Thank you for your valuable feedback.
Comment 7: A final comment concerns citations. Please standardise the naming of citations in a scientific article.
Response: We appreciate the reviewer’s comment regarding the standardisation of citation naming in the manuscript. In response, we have carefully reviewed all citations and ensured that they are consistently formatted throughout the document. Reference management software was used therefore these changes have not been tracked. Thank you for your valuable feedback.
Round 2
Reviewer 1 Report
Comments and Suggestions for Authors
The manuscript can be considered for publication in its current form.